# Fast Multivariate Spatio-temporal Analysis via Low Rank Tensor Learning

**Mohammad Taha Bahadori***
Dept. of Electrical Engineering
Univ. of Southern California
Los Angeles, CA 90089
mohammab@usc.edu

**Qi (Rose) Yu***
Dept. of Computer Science
Univ. of Southern California
Los Angeles, CA 90089
qiyu@usc.edu

**Yan Liu**
Dept. of Computer Science
Univ. of Southern California
Los Angeles, CA 90089
yanliu.cs@usc.edu

## Abstract

Accurate and efficient analysis of multivariate spatio-temporal data is critical in climatology, geology, and sociology applications. Existing models usually assume simple inter-dependence among variables, space, and time, and are computationally expensive. We propose a unified low rank tensor learning framework for multivariate spatio-temporal analysis, which can conveniently incorporate different properties in spatio-temporal data, such as spatial clustering and shared structure among variables. We demonstrate how the general framework can be applied to cokriging and forecasting tasks, and develop an efficient greedy algorithm to solve the resulting optimization problem with convergence guarantee. We conduct experiments on both synthetic datasets and real application datasets to demonstrate that our method is not only significantly faster than existing methods but also achieves lower estimation error.

## 1 Introduction

Spatio-temporal data provide unique information regarding "where" and "when", which is essential to answer many important questions in scientific studies from geology, climatology to sociology. In the context of big data, we are confronted with a series of new challenges when analyzing spatio-temporal data because of the complex spatial and temporal dependencies involved.

A plethora of excellent work has been conducted to address the challenge and achieved successes to a certain extent [8, 13]. Often times, geostatistical models use cross variogram and cross covariance functions to describe the intrinsic dependency structure. However, the parametric form of cross variogram and cross covariance functions impose strong assumptions on the spatial and temporal correlation, which requires domain knowledge and manual work. Furthermore, parameter learning of those statistical models is computationally expensive, making them infeasible for large-scale applications.

Cokriging and forecasting are two central tasks in multivariate spatio-temporal analysis. Cokriging utilizes the spatial correlations to predict the value of the variables for new locations. One widely adopted method is multitask Gaussian process (MTGP) [4], which assumes a Gaussian process prior over latent functions to directly induce correlations between tasks. However, for a cokriging task with $M$ variables of $P$ locations for $T$ time stamps, the time complexity of MTGP is $\mathcal{O}(M^3 P^3 T)$ [4]. For forecasting, popular methods in multivariate time series analysis include vector autoregressive (VAR) models, autoregressive integrated moving average (ARIMA) models, and cointegration models. An alternative method for spatio-temporal analysis is Bayesian hierarchical spatio-temporal models with either separable and non-separable space-time covariance functions [6]. Rank reduced

models have been proposed to capture the inter-dependency among variables [1]. However, very few models can directly handle the correlations among variables, space and time simultaneously in a scalable way. In this paper, we aim to address this problem by presenting a unified framework for many spatio-temporal analysis tasks that are scalable for large-scale applications.

Tensor representation provides a convenient way to capture inter-dependencies along multiple dimensions. Therefore it is natural to represent the multivariate spatio-temporal data in tensor. Recent advances in low rank learning have led to simple models that can capture the commonalities among each mode of the tensor [15, 20]. Similar argument can be found in the literature of spatial data recovery [11], neuroimaging analysis [26], and multi-task learning [20]. Our work builds upon recent advances in low rank tensor learning [15, 11, 26] and further considers the scenario where additional side information of data is available. For example, in geo-spatial applications, apart from measurements of multiple variables, geographical information is available to infer location adjacency; in social network applications, friendship network structure is collected to obtain preference similarity. To utilize the side information, we can construct a Laplacian regularizer from the similarity matrices, which favors locally smooth solutions.

We develop a fast greedy algorithm for learning low rank tensors based on the greedy structure learning framework [2, 24, 21]. Greedy low rank tensor learning is efficient, as it does not require full singular value decomposition of large matrices as opposed to other alternating direction methods [11]. We also provide a bound on the difference between the loss function at our greedy solution and the one at the globally optimal solution. Finally, we present experiment results on simulation datasets as well as application datasets in climate and social network analysis, which show that our algorithm is faster and achieves higher prediction accuracy than state-of-art approaches in cokriging and forecasting tasks.

## 2 Tensor formulation for multivariate spatio-temporal analysis

The critical element in multivariate spatio-temporal analysis is an efficient way to incorporate the spatial temporal correlations into modeling and automatically capture the shared structures across variables, locations, and time. In this section, we present a unified low rank tensor learning framework that can perform various types of spatio-temporal analysis. We will use two important applications, i.e., cokriging and forecasting, to motivate and describe the framework.

### 2.1 Cokriging

In geostatistics, cokriging is the task of interpolating the data of one variable for unknown locations by taking advantage of the observations of variables from known locations. For example, by making use of the correlations between precipitation and temperature, we can obtain more precise estimate of temperature in unknown locations than univariate kriging. Formally, denote the complete data for $P$ locations over $T$ time stamps with $M$ variables as $\mathcal{X} \in \mathbb{R}^{P \times T \times M}$. We only observe the measurements for a subset of locations $\Omega \subset \{1, \ldots, P\}$ and their side information such as longitude and latitude. Given the measurements $\mathcal{X}_\Omega$ and the side information, the goal is to estimate a tensor $\mathcal{W} \in \mathbb{R}^{P \times T \times M}$ that satisfies $\mathcal{W}_\Omega = \mathcal{X}_\Omega$. Here $\mathcal{X}_\Omega$ represents the outcome of applying the index operator $I_\Omega$ to $\mathcal{X}_{:,:,m}$ for all variables $m = 1, \ldots, M$. The index operator $I_\Omega$ is a diagonal matrix whose entries are one for the locations included in $\Omega$ and zero otherwise.

Two key consistency principles have been identified for effective cokriging [9, Chapter 6.2]: (1) Global consistency: the data on the same structure are likely to be similar. (2) Local consistency: the data in close locations are likely to be similar. The former principle is akin to the *cluster assumption* in semi-supervised learning [25]. We incorporate these principles in a concise and computationally efficient low-rank tensor learning framework.

To achieve global consistency, we constrain the tensor $\mathcal{W}$ to be low rank. The low rank assumption is based on the belief that high correlations exist within variables, locations and time, which leads to natural clustering of the data. Existing literature have explored the low rank structure among these three dimensions separately, e.g., multi-task learning [19] for variable correlation, fixed rank kriging [7] for spatial correlations. Low rankness assumes that the observed data can be described with a few latent factors. It enforces the commonalities along three dimensions without an explicit form for the shared structures in each dimension.

For local consistency, we construct a regularizer via the spatial Laplacian matrix. The Laplacian matrix is defined as $L = D - A$, where $A$ is a kernel matrix constructed by pairwise similarity and diagonal matrix $D_{i,i} = \sum_j (A_{i,j})$. Similar ideas have been explored in matrix completion [16]. In cokriging literature, the local consistency is enforced via the spatial covariance matrix. The Bayesian models often impose the Gaussian process prior on the observations with the covariance matrix $K = K_v \otimes K_x$ where $K_v$ is the covariance between variables and $K_x$ is that for locations. The Laplacian regularization term corresponds to the relational Gaussian process [5] where the covariance matrix is approximated by the spatial Laplacian.

In summary, we can perform cokriging and find the value of tensor $\mathcal{W}$ by solving the following optimization problem:

$$\widehat{\mathcal{W}} = \underset{\mathcal{W}}{\operatorname{argmin}} \left\{ \|\mathcal{W}_\Omega - \mathcal{X}_\Omega\|_F^2 + \mu \sum_{m=1}^{M} \operatorname{tr}(\mathcal{W}_{:,:,m}^\top L \mathcal{W}_{:,:,m}) \right\} \quad \text{s.t.} \quad \operatorname{rank}(\mathcal{W}) \leq \rho, \quad (1)$$

where the Frobenius norm of a tensor $\mathcal{A}$ is defined as $\|\mathcal{A}\|_F = \sqrt{\sum_{i,j,k} \mathcal{A}_{i,j,k}^2}$ and $\mu, \rho > 0$ are the parameters that make tradeoff between the local and global consistency, respectively. The low rank constraint finds the principal components of the tensor and reduces the complexity of the model while the Laplacian regularizer clusters the data using the relational information among the locations. By learning the right tradeoff between these two techniques, our method is able to benefit from both. Due to the various definitions of tensor rank, we use *rank* as supposition for rank complexity, which will be specified in later section.

## 2.2 Forecasting

Forecasting estimates the future value of multivariate time series given historical observations. For ease of presentation, we use the classical VAR model with $K$ lags and coefficient tensor $\mathcal{W} \in \mathbb{R}^{P \times KP \times M}$ as an example. Using the matrix representation, the VAR($K$) process defines the following data generation process:

$$\mathcal{X}_{:,t,m} = \mathcal{W}_{:,:,m} \mathbf{X}_{t,m} + \mathcal{E}_{:,t,m}, \quad \text{for } m = 1, \ldots, M \text{ and } t = K+1, \ldots, T, \quad (2)$$

where $\mathbf{X}_{t,m} = [\mathcal{X}_{:,t-1,m}^\top, \ldots, \mathcal{X}_{:,t-K,m}^\top]^\top$ denotes the concatenation of $K$-lag historical data before time $t$. The noise tensor $\mathcal{E}$ is a multivariate Gaussian with zero mean and unit variance .

Existing multivariate regression methods designed to capture the complex correlations, such as Tucker decomposition [20], are computationally expensive. A scalable solution requires a simpler model that also efficiently accounts for the shared structures in variables, space, and time. Similar global and local consistency principles still hold in forecasting. For global consistency, we can use low rank constraint to capture the commonalities of the variables as well as the spatial correlations on the model parameter tensor, as in [8]. For local consistency, we enforce the predicted value for close locations to be similar via spatial Laplacian regularization. Thus, we can formulate the forecasting task as the following optimization problem over the model coefficient tensor $\mathcal{W}$:

$$\widehat{\mathcal{W}} = \underset{\mathcal{W}}{\operatorname{argmin}} \left\{ \|\widehat{\mathcal{X}} - \mathcal{X}\|_F^2 + \mu \sum_{m=1}^{M} \operatorname{tr}(\widehat{\mathcal{X}}_{:,:,m}^\top L \widehat{\mathcal{X}}_{:,:,m}) \right\} \quad \text{s.t.} \quad \operatorname{rank}(\mathcal{W}) \leq \rho, \ \widehat{\mathcal{X}}_{:,t,m} = \mathcal{W}_{:,:,m} \mathbf{X}_{t,m}$$

$$(3)$$

Though cokriging and forecasting are two different tasks, we can easily see that both formulations follow the global and local consistency principles and can capture the inter-correlations from spatial-temporal data.

## 2.3 Unified Framework

We now show that both cokriging and forecasting can be formulated into the same tensor learning framework. Let us rewrite the loss function in Eq. (1) and Eq. (3) in the form of multitask regression and complete the quadratic form for the loss function. The cokriging task can be reformulated as follows:

$$\widehat{\mathcal{W}} = \underset{\mathcal{W}}{\operatorname{argmin}} \left\{ \sum_{m=1}^{M} \|\mathcal{W}_{:,:,m} H - (H^\top)^{-1} \mathcal{X}_{\Omega,m}\|_F^2 \right\} \quad \text{s.t.} \quad \operatorname{rank}(\mathcal{W}) \leq \rho \quad (4)$$

where we define $HH^\top = I_\Omega + \mu L$.[1] For the forecasting problem, $HH^\top = I_P + \mu L$ and we have:

$$\widehat{\mathcal{W}} = \underset{\mathcal{W}}{\operatorname{argmin}} \left\{ \sum_{m=1}^{M} \sum_{t=K+1}^{T} \|H\mathcal{W}_{:,:,m}\mathbf{X}_{t,m} - (H^{-1})\mathcal{X}_{:,t,m}\|_F^2 \right\} \quad \text{s.t.} \quad \operatorname{rank}(\mathcal{W}) \leq \rho, \quad (5)$$

By slight change of notation (cf. Appendix D), we can easily see that the optimal solution of both problems can be obtained by the following optimization problem with appropriate choice of tensors $\mathcal{Y}$ and $\mathcal{V}$:

$$\widehat{\mathcal{W}} = \underset{\mathcal{W}}{\operatorname{argmin}} \left\{ \sum_{m=1}^{M} \|\mathcal{W}_{:,:,m}\mathcal{Y}_{:,:,m} - \mathcal{V}_{:,:,m}\|_F^2 \right\} \quad \text{s.t.} \quad \operatorname{rank}(\mathcal{W}) \leq \rho. \quad (6)$$

After unifying the objective function, we note that tensor rank has different notions such as CP rank, Tucker rank and mode n-rank [15, 11]. In this paper, we choose the mode-n rank, which is computationally more tractable [11, 23]. The mode-n rank of a tensor $\mathcal{W}$ is the rank of its mode-n unfolding $\mathcal{W}_{(n)}$.[2] In particular, for a tensor $\mathcal{W}$ with $N$ mode, we have the following definition:

$$\text{mode-n rank}(\mathcal{W}) = \sum_{n=1}^{N} \operatorname{rank}(\mathcal{W}_{(n)}). \quad (7)$$

A common practice to solve this formulation with mode $n$-rank constraint is to relax the rank constraint to a convex nuclear norm constraint [11, 23]. However, those methods are computationally expensive since they need full singular value decomposition of large matrices. In the next section, we present a fast greedy algorithm to tackle the problem.

## 3   Fast greedy low rank tensor learning

To solve the non-convex problem in Eq. (6) and find its optimal solution, we propose a greedy learning algorithm by successively adding rank-1 estimation of the mode-n unfolding. The main idea of the algorithm is to unfold the tensor into a matrix, seek for its rank-1 approximation and then fold back into a tensor with same dimensionality. We describe this algorithm in three steps: (i) First, we show that we can learn rank-1 matrix estimations efficiently by solving a generalized eigenvalue problem, (ii) We use the rank-1 matrix estimation to greedily solve the original tensor rank constrained problem, and (iii) We propose an enhancement via orthogonal projections after each greedy step.

**Optimal rank-1 Matrix Learning**   The following lemma enables us to find such optimal rank-1 estimation of the matrices.

**Lemma 1.** *Consider the following rank constrained problem:*

$$\widehat{A}_1 = \underset{A:\operatorname{rank}(A)=1}{\operatorname{argmin}} \left\{ \|Y - AX\|_F^2 \right\}, \quad (8)$$

*where $Y \in \mathbb{R}^{q \times n}$, $X \in \mathbb{R}^{p \times n}$, and $A \in \mathbb{R}^{q \times p}$. The optimal solution of $\widehat{A}_1$ can be written as $\widehat{A}_1 = \widehat{\mathbf{u}}\widehat{\mathbf{v}}^\top$, $\|\widehat{\mathbf{v}}\|_2 = 1$ where $\widehat{\mathbf{v}}$ is the dominant eigenvector of the following generalized eigenvalue problem:*

$$(XY^\top Y X^\top)\mathbf{v} = \lambda(XX^\top)\mathbf{v} \quad (9)$$

*and $\widehat{\mathbf{u}}$ can be computed as*

$$\widehat{\mathbf{u}} = \frac{1}{\widehat{\mathbf{v}}^\top XX^\top \widehat{\mathbf{v}}} YX^\top \widehat{\mathbf{v}}. \quad (10)$$

Proof is deferred to Appendix A. Eq. (9) is a generalized eigenvalue problem whose dominant eigenvector can be found efficiently [12]. If $XX^\top$ is full rank, as assumed in Theorem 2, the problem is simplified to a regular eigenvalue problem whose dominant eigenvector can be efficiently computed.

**Algorithm 1** Greedy Low-rank Tensor Learning

1: **Input:** transformed data $\mathcal{Y}, \mathcal{V}$ of $M$ variables, stopping criteria $\eta$
2: **Output:** $N$ mode tensor $\mathcal{W}$
3: Initialize $\mathcal{W} \leftarrow 0$
4: **repeat**
5:     **for** $n = 1$ **to** $N$ **do**
6:         $B_n \leftarrow \underset{B:\, \text{rank}(B)=1}{\text{argmin}} \mathcal{L}(\text{refold}(\mathcal{W}_{(n)} + B); \mathcal{Y}, \mathcal{V})$
7:         $\Delta_n \leftarrow \mathcal{L}(\mathcal{W}; \mathcal{Y}, \mathcal{V}) - \mathcal{L}(\text{refold}(\mathcal{W}_{(n)} + B_n); \mathcal{Y}, \mathcal{V})$
8:     **end for**
9:     $n^* \leftarrow \underset{n}{\text{argmax}}\{\Delta_n\}$
10:    **if** $\Delta_{n^*} > \eta$ **then**
11:       $\mathcal{W} \leftarrow \mathcal{W} + \text{refold}(B_{n^*}, n^*)$
12:    **end if**
13:    $\mathcal{W} \leftarrow \text{argmin}_{\substack{\text{row}(\mathcal{A}_{(1)}) \subseteq \text{row}(\mathcal{W}_{(1)}) \\ \text{col}(\mathcal{A}_{(1)}) \subseteq \text{col}(\mathcal{W}_{(1)})}} \mathcal{L}(\mathcal{A}; \mathcal{Y}, \mathcal{V})$       *# Optional Orthogonal Projection Step.*
14: **until** $\Delta_{n^*} < \eta$

**Greedy Low n-rank Tensor Learning** The optimal rank-1 matrix learning serves as a basic element in our greedy algorithm. Using Lemma 1, we can solve the problem in Eq. (6) in the *Forward Greedy Selection* framework as follows: at each iteration of the greedy algorithm, it searches for the mode that gives the largest decrease in the objective function. It does so by unfolding the tensor in that mode and finding the best rank-1 estimation of the unfolded tensor. After finding the optimal mode, it adds the rank-1 estimate in that mode to the current estimation of the tensor. Algorithm 1 shows the details of this approach, where $\mathcal{L}(\mathcal{W}; \mathcal{Y}, \mathcal{V}) = \sum_{m=1}^{M} \|\mathcal{W}_{:,:,m} \mathcal{Y}_{:,:,m} - \mathcal{V}_{:,:,m}\|_F^2$. Note that we can find the optimal rank-1 solution in only one of the modes, but it is enough to guarantee the convergence of our greedy algorithm.

Theorem 2 bounds the difference between the loss function evaluated at each iteration of the greedy algorithm and the one at the globally optimal solution.

**Theorem 2.** *Suppose in Eq. (6) the matrices $\mathcal{Y}_{:,:,m}^\top \mathcal{Y}_{:,:,m}$ for $m = 1, \ldots, M$ are positive definite. The solution of Algo. 1 at its $k$th iteration step satisfies the following inequality:*

$$\mathcal{L}(\mathcal{W}_k; \mathcal{Y}, \mathcal{V}) - \mathcal{L}(\mathcal{W}^*; \mathcal{Y}, \mathcal{V}) \leq \frac{(\|\mathcal{Y}\|_2 \|\mathcal{W}_{(1)}^*\|_*)^2}{(k+1)}, \tag{11}$$

*where $\mathcal{W}^*$ is the global minimizer of the problem in Eq. (6) and $\|\mathcal{Y}\|_2$ is the largest singular value of a block diagonal matrix created by placing the matrices $\mathcal{Y}(:,:,m)$ on its diagonal blocks.*

The detailed proof is given in Appendix B. The key idea of the proof is that the amount of decrease in the loss function by each step in the selected mode is not smaller than the amount of decrease if we had selected the first mode. The theorem shows that we can obtain the same rate of convergence for learning low rank tensors as achieved in [22] for learning low rank matrices. The greedy algorithm in Algorithm 1 is also connected to mixture regularization in [23]: the mixture approach decomposes the solution into a set of low rank structures while the greedy algorithm successively learns a set of rank one components.

**Greedy Algorithm with Orthogonal Projections** It is well-known that the forward greedy algorithm may make steps in sub-optimal directions because of noise. A common solution to alleviate the effect of noise is to make orthogonal projections after each greedy step [2, 21]. Thus, we enhance the forward greedy algorithm by projecting the solution into the space spanned by the singular vectors of its mode-1 unfolding. The greedy algorithm with *orthogonal* projections performs an extra step in line 13 of Algorithm 1: It finds the top $k$ singular vectors of the solution: $[U, S, V] \leftarrow \text{svd}(\mathcal{W}_{(1)}, k)$ where $k$ is the iteration number. Then it finds the best solution in the space spanned by $U$ and $V$ by solving $\widehat{S} \leftarrow \min_S \mathcal{L}(USV^\top, \mathcal{Y}, \mathcal{V})$ which has a closed form solution. Finally, it reconstructs the solution: $\mathcal{W} \leftarrow \text{refold}(U\widehat{S}V^\top, 1)$. Note that the projection only needs to find top $k$ singular vectors which can be computed efficiently for small values of $k$.

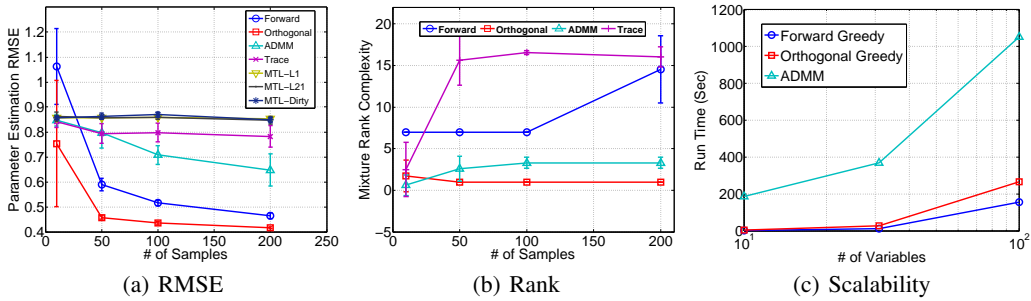

(a) RMSE        (b) Rank        (c) Scalability

Figure 1: Tensor estimation performance comparison on the synthetic dataset over 10 random runs. (a) parameter Estimation RMSE with training time series length, (b) Mixture Rank Complexity with training time series length, (c) running time for one single round with respect to number of variables.

## 4   Experiments

We evaluate the efficacy of our algorithms on synthetic datasets and real-world application datasets.

### 4.1   Low rank tensor learning on synthetic data

For empirical evaluation, we compare our method with multitask learning (MTL) algorithms, which also utilize the commonalities between different prediction tasks for better performance. We use the following baselines: (1) Trace norm regularized MTL (*Trace*), which seeks the low rank structure only on the task dimension; (2) Multilinear MTL [20], which adapts the convex relaxation of low rank tensor learning solved with Alternating Direction Methods of Multiplier (*ADMM*) [10] and Tucker decomposition to describe the low rankness in multiple dimensions; (3) *MTL-$L_1$* , *MTL-$L_{21}$* [19], and *MTL-$L_{\mathrm{Dirty}}$* [14], which investigate joint sparsity of the tasks with $L_p$ norm regularization. For MTL-$L_1$ , MTL-$L_{21}$ [19] and MTL-$L_{\mathrm{Dirty}}$, we use MALSAR Version 1.1 [27].

We construct a model coefficient tensor $\mathcal{W}$ of size $20 \times 20 \times 10$ with CP rank equals to 1. Then, we generate the observations $\mathcal{Y}$ and $\mathcal{V}$ according to multivariate regression model $\mathcal{V}_{:,:,m} = \mathcal{W}_{:,:,m}\mathcal{Y}_{:,:,m} + \mathcal{E}_{:,:,m}$ for $m = 1, \ldots, M$, where $\mathcal{E}$ is tensor with zero mean Gaussian noise elements. We split the synthesized data into training and testing time series and vary the length of the training time series from 10 to 200. For each training length setting, we repeat the experiments for 10 times and select the model parameters via 5-fold cross validation. We measure the prediction performance via two criteria: parameter estimation accuracy and rank complexity. For accuracy, we calculate the RMSE of the estimation versus the true model coefficient tensor. For rank complexity, we calculate the mixture rank complexity [23] as $MRC = \frac{1}{n} \sum_{n=1}^{N} \mathrm{rank}(\mathcal{W}_{(n)})$.

The results are shown in Figure 1(a) and 1(b). We omit the Tucker decomposition as the results are not comparable. We can clearly see that the proposed greedy algorithm with orthogonal projections achieves the most accurate tensor estimation. In terms of rank complexity, we make two observations: (i) Given that the tensor CP rank is 1, greedy algorithm with orthogonal projections produces the estimate with the lowest rank complexity. This can be attributed to the fact that the orthogonal projections eliminate the redundant rank-1 components that fall in the same spanned space. (ii) The rank complexity of the forward greedy algorithm increases as we enlarge the sample size. We believe that when there is a limited number of observations, most of the new rank-1 elements added to the estimate are not accurate and the cross-validation steps prevent them from being added to the model. However, as the sample size grows, the rank-1 estimates become more accurate and they are preserved during the cross-validation.

To showcase the scalability of our algorithm, we vary the number of variables and generate a series of tensor $\mathcal{W} \in \mathbb{R}^{20 \times 20 \times M}$ for M from 10 to 100 and record the running time (in seconds) for three tensor learning algorithms, i.e, forward greedy, greedy with orthogonal projections and ADMM. We measure the run time on a machine with a 6-core 12-thread Intel Xenon 2.67GHz processor and 12GB memory. The results are shown in Figure 1(c). The running time of ADMM increase rapidly with the data size while the greedy algorithm stays steady, which confirms the speedup advantage of the greedy algorithm.

Table 1: Cokriging RMSE of 6 methods averaged over 10 runs. In each run, 10% of the locations are assumed missing.

| DATASET | ADMM | FORWARD | ORTHOGONAL | SIMPLE | ORDINARY | MTGP |
|---------|------|---------|------------|--------|----------|------|
| USHCN | 0.8051 | 0.7594 | **0.7210** | 0.8760 | 0.7803 | 1.0007 |
| CCDS | 0.8292 | 0.5555 | **0.4532** | 0.7634 | 0.7312 | 1.0296 |
| YELP | 0.7730 | 0.6993 | **0.6958** | NA | NA | NA |
| FOURSQUARE | 0.1373 | 0.1338 | **0.1334** | NA | NA | NA |

## 4.2 Spatio-temporal analysis on real world data

We conduct cokriging and forecasting experiments on four real-world datasets:

**USHCN** The U.S. Historical Climatology Network Monthly (USHCN)[3] dataset consists of monthly climatological data of 108 stations spanning from year 1915 to 2000. It has three climate variables: (1) daily maximum, (2) minimum temperature averaged over month, and (3) total monthly precipitation.

**CCDS** The Comprehensive Climate Dataset (CCDS)[4] is a collection of climate records of North America from [18]. The dataset was collected and pre-processed by five federal agencies. It contains monthly observations of 17 variables such as Carbon dioxide and temperature spanning from 1990 to 2001. The observations were interpolated on a $2.5 \times 2.5$ degree grid, with 125 observation locations.

**Yelp** The Yelp dataset[5] contains the user rating records for 22 categories of businesses on Yelp over ten years. The processed dataset includes the rating values (1-5) binned into 500 time intervals and the corresponding social graph for 137 active users. The dataset is used for the spatio-temporal recommendation task to predict the missing user ratings across all business categories.

**Foursquare** The Foursquare dataset [17] contains the users' check-in records in Pittsburgh area from Feb 24 to May 23, 2012, categorized by different venue types such as Art & Entertainment, College & University, and Food. The dataset records the number of check-ins by 121 users in each of the 15 category of venues over 1200 time intervals, as well as their friendship network.

### 4.2.1 Cokriging

We compare the cokriging performance of our proposed method with the classical cokriging approaches including simple kriging and ordinary cokriging with nonbias condition [13] which are applied to each variables separately. We further compare with multitask Gaussian process (MTGP) [4] which also considers the correlation among variables. We also adapt ADMM for solving the nuclear norm relaxed formulation of the cokriging formulation as a baseline (see Appendix C for more details). For USHCN and CCDS, we construct a Laplacian matrix by calculating the pairwise Haversine distance of locations. For Foursquare and Yelp, we construct the graph Laplacian from the user friendship network.

For each dataset, we first normalize it by removing the trend and diving by the standard deviation. Then we randomly pick 10% of locations (or users for Foursquare) and eliminate the measurements of all variables over the whole time span. Then, we produce the estimates for all variables of each timestamp. We repeat the procedure for 10 times and report the average prediction RMSE for all timestamps and 10 random sets of missing locations. We use the MATLAB Kriging Toolbox[6] for the classical cokriging algorithms and the MTGP code provided by [4].

Table 1 shows the results for the cokriging task. The greedy algorithm with orthogonal projections is significantly more accurate in all three datasets. The baseline cokriging methods can only handle the two dimensional longitude and latitude information, thus are not applicable to the Foursquare and Yelp dataset with additional friendship information. The superior performance of the greedy algorithm can be attributed to two of its properties: (1) It can obtain low rank models and achieve global consistency; (2) It usually has lower estimation bias compared to nuclear norm relaxed methods.

Table 2: Forecasting RMSE for VAR process with 3 lags, trained with 90% of the time series.

| Dataset | Tucker | ADMM | Forward | Ortho | OrthoNL | Trace | MTL$_{l1}$ | MTL$_{l21}$ | MTL$_{dirty}$ |
|---|---|---|---|---|---|---|---|---|---|
| USHCN | **0.8975** | 0.9227 | 0.9171 | 0.9069 | 0.9175 | 0.9273 | 0.9528 | 0.9543 | 0.9735 |
| CCDS | 0.9438 | 0.8448 | 0.8810 | **0.8325** | 0.8555 | 0.8632 | 0.9105 | 0.9171 | 1.0950 |
| FSQ | 0.1492 | 0.1407 | 0.1241 | **0.1223** | 0.1234 | 0.1245 | 0.1495 | 0.1495 | 0.1504 |

Table 3: Running time (in seconds) for cokriging and forecasting.

| | Cokriging | | | | Forecasting | | |
|---|---|---|---|---|---|---|---|
| Dataset | USHCN | CCDS | YELP | FSQ | USHCN | CCDS | FSQ |
| ORTHO | 93.03 | 16.98 | 78.47 | 91.51 | 75.47 | 21.38 | 37.70 |
| ADMM | 791.25 | 320.77 | 2928.37 | 720.40 | 235.73 | 45.62 | 33.83 |

### 4.2.2 Forecasting

We present the empirical evaluation on the forecasting task by comparing with multitask regression algorithms. We split the data along the temporal dimension into 90% training set and 10% testing set. We choose VAR(3) model and during the training phase, we use 5-fold cross-validation.

As shown in Table 2, the greedy algorithm with orthogonal projections again achieves the best prediction accuracy. Different from the cokriging task, forecasting does not necessarily need the correlations of locations for prediction. One might raise the question as to whether the Laplacian regularizer helps. Therefore, we report the results for our formulation without Laplacian (ORTHONL) for comparison. For efficiency, we report the running time (in seconds) in Table 3 for both tasks of cokriging and forecasting. Compared with ADMM, which is a competitive baseline also capturing the commonalities among variables, space, and time, our greedy algorithm is much faster for most datasets.

As a qualitative study, we plot the map of most predictive regions analyzed by the greedy algorithm using CCDS dataset in Fig. 2. Based on the concept of how informative the past values of the climate measurements in a specific location are in predicting future values of other time series, we define the aggregate strength of predictiveness of each region as $w(t) = \sum_{p=1}^{P} \sum_{m=1}^{M} |\mathcal{W}_{p,t,m}|$. We can see that two regions are identified as the most predictive regions: (1) The southwest region, which reflects the impact of the Pacific ocean and (2) The southeast region, which frequently experiences relative sea level rise, hurricanes, and storm surge in Gulf of Mexico. Another interesting region lies in the center of Colorado, where the Rocky mountain valleys act as a funnel for the winds from the west, providing locally divergent wind patterns.

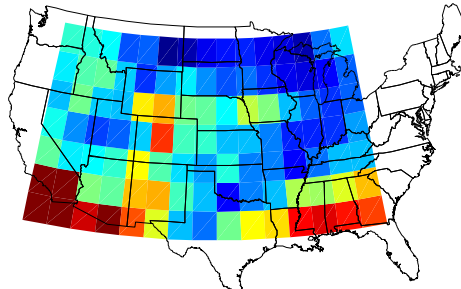

Figure 2: Map of most predictive regions analyzed by the greedy algorithm using 17 variables of the CCDS dataset. Red color means high predictiveness whereas blue denotes low predictiveness.

## 5 Conclusion

In this paper, we study the problem of multivariate spatio-temporal data analysis with an emphasis on two tasks: cokriging and forecasting. We formulate the problem into a general low rank tensor learning framework which captures both the global consistency and the local consistency principle. We develop a fast and accurate greedy solver with theoretical guarantees for its convergence. We validate the correctness and efficiency of our proposed method on both the synthetic dataset and real-application datasets. For future work, we are interested in investigating different forms of shared structure and extending the framework to capture non-linear correlations in the data.

### Acknowledgment

We thank the anonymous reviewers for their helpful feedback and comments. The research was sponsored by the NSF research grants IIS-1134990, IIS- 1254206 and Okawa Foundation Research Award. The views and conclusions are those of the authors and should not be interpreted as representing the official policies of the funding agency, or the U.S. Government.

## Footnotes

*Authors have equal contributions.

[1] We can use Cholesky decomposition to obtain $H$. In the rare cases that $I_\Omega + \mu L$ is not full rank, $\epsilon I_P$ is added where $\epsilon$ is a very small positive value.

[2] The mode-$n$ unfolding of a tensor is the matrix resulting from treating $n$ as the first mode of the matrix, and cyclically concatenating other modes. Tensor refolding is the reverse direction operation [15].

[3] http://www.ncdc.noaa.gov/oa/climate/research/ushcn

[4] http://www-bcf.usc.edu/~liu32/data/NA-1990-2002-Monthly.csv

[5] http://www.yelp.com/dataset_challenge

[6] http://globec.whoi.edu/software/kriging/V3/english.html

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
