[Supplementary Material · supp.pdf]

# A   Proof of Lemma 1

*Proof.* The original problem has the following form:

$$\widehat{A} = \underset{A:\mathrm{rank}(A)=1}{\mathrm{argmin}} \left\{ \|Y - AX\|_F^2 \right\} \tag{12}$$

We can rewrite the optimization problem in Eq. (12) as estimation of $\alpha \in \mathbb{R}$, $\mathbf{u} \in \mathbb{R}^{q \times 1}, \|\mathbf{u}\|_2 = 1$, and $\mathbf{v} \in \mathbb{R}^{p \times 1}, \|\mathbf{v}\|_2 = 1$ such that:

$$\widehat{\alpha}, \widehat{\mathbf{u}}, \widehat{\mathbf{v}} = \underset{\alpha,\mathbf{u},\mathbf{v}:\|\mathbf{u}\|_2=1,\|\mathbf{v}\|_2=1}{\mathrm{argmin}} \left\{ \left\|Y - \alpha\mathbf{u}\mathbf{v}^\top X\right\|_F^2 \right\} \tag{13}$$

We will minimize the above objective function in three steps: First, minimization in terms of $\alpha$ yields $\widehat{\alpha} = \langle Y, \mathbf{u}\mathbf{v}^\top X\rangle / \|\mathbf{u}\mathbf{v}^\top X\|_F^2$, where we have assumed that $\mathbf{v}^\top X \neq \mathbf{0}$. Hence, we have:

$$\widehat{\mathbf{u}}, \widehat{\mathbf{v}} = \underset{\mathbf{u},\mathbf{v}:\|\mathbf{u}\|_2=1,\|\mathbf{v}\|_2=1}{\mathrm{argmax}} \frac{\mathrm{tr}((\mathbf{u}\mathbf{v}^\top X)^\top Y)^2}{\|\mathbf{u}\mathbf{v}^\top X\|_F^2} \tag{14}$$

The objective function can be rewritten $\mathrm{tr}\left\{(\mathbf{u}\mathbf{v}^\top X)^\top Y\right\} = \mathrm{tr}\left\{X^\top \mathbf{v}\mathbf{u}^\top Y\right\} = \mathrm{tr}\left\{YX^\top \mathbf{v}\mathbf{u}^\top\right\}$. Some algebra work on the denominator yields $\|\mathbf{u}\mathbf{v}^\top X\|_F^2 = \mathrm{tr}\left\{(\mathbf{u}\mathbf{v}^\top X)^\top (\mathbf{u}\mathbf{v}^\top X)\right\} = \mathrm{tr}\left\{X^\top \mathbf{v}\mathbf{u}^\top \mathbf{u}\mathbf{v}^\top X\right\} = \mathrm{tr}\left\{X^\top \mathbf{v}\mathbf{v}^\top X\right\} = \mathbf{v}^\top XX^\top \mathbf{v}$. This implies that the denominator is independent of $\mathbf{u}$ and the optimal value of $\mathbf{u}$ in Eq. (14) is proportional to $YX^\top \mathbf{v}$. Hence, we need to first find the optimal value of $\mathbf{v}$ and then compute $\mathbf{u} = (YX^\top \mathbf{v})/\|YX^\top \mathbf{v}\|_2$. Substitution of the optimal value of $\mathbf{u}$ yields:

$$\widehat{\mathbf{v}} = \underset{\mathbf{v}:\|\mathbf{v}\|_2=1}{\mathrm{argmax}} \frac{\mathbf{v}^\top XY^\top YX^\top \mathbf{v}}{\mathbf{v}^\top XX^\top \mathbf{v}} \tag{15}$$

Note that the objective function is bounded and invariant of $\|\mathbf{v}\|_2$, hence the $\|\mathbf{v}\|_2 = 1$ constraint can be relaxed. Now, suppose the value of $\mathbf{v}^\top XX^\top \mathbf{v}$ for optimal choice of vectors $\mathbf{v}$ is $t$. We can rewrite the optimization in Eq. (15) as

$$\widehat{\mathbf{v}} = \underset{\mathbf{v}}{\mathrm{argmax}} \ \mathbf{v}^\top XY^\top YX^\top \mathbf{v}$$
$$\text{s.t.} \qquad \mathbf{v}^\top XX^\top \mathbf{v} = t \tag{16}$$

Using the Lagrangian multipliers method, we can show that there is a value for $\lambda$ such that the solution $\widehat{\mathbf{v}}$ for the dual problem is the optimal solution for Eq. (16). Hence, we need to solve the following optimization problem for $\mathbf{v}$:

$$\widehat{\mathbf{v}} = \underset{\mathbf{v}:\|\mathbf{v}\|_2=1}{\mathrm{argmax}} \left\{\mathbf{v}^\top XY^\top YX^\top \mathbf{v} - \lambda\mathbf{v}^\top XX^\top \mathbf{v}\right\}$$
$$= \underset{\mathbf{v}:\|\mathbf{v}\|_2=1}{\mathrm{argmax}} \left\{\mathbf{v}^\top X(Y^\top Y - \lambda I)X^\top \mathbf{v}\right\} \tag{17}$$

Eq. (17) implies that $\mathbf{v}$ is the dominant eigenvector of $X(Y^\top Y - \lambda I)X^\top$. Hence, we are able to find the optimal value of both $\mathbf{u}$ and $\mathbf{v}$ for the given value of $\lambda$. For simplicity of notation, let's define $P \triangleq XX^\top$ and $Q \triangleq XY^\top YX^\top$. Consider the equations obtained by solving the Lagrangian dual of Eq. (16):

$$Q\mathbf{v} = \lambda P\mathbf{v} \tag{18}$$
$$\|\mathbf{v}^\top X\|_2^2 = t, \tag{19}$$
$$\lambda \geq 0. \tag{20}$$

Eq. (18) describes a generalized positive definite eigenvalue problem. Hence, we can select $\lambda_{\max} = \lambda_1(Q, P)$ which maximizes the objective function in Eq. (15). The optimal value of $\mathbf{u}$ can be found by substitution of optimal $\mathbf{v}$ in Eq. (14) and simple algebra yields the result in Lemma 1. $\qquad\square$

# B  Proof of Theorem 2

Note that intuitively, since our greedy steps are optimal in the first mode, we can see that our bound should be at least as tight as the bound of [21]. Here is the formal proof of Theorem 2.

*Proof.* Let's denote the loss function at $k^{th}$ step by

$$\mathcal{L}(\mathcal{Y}, \mathcal{V}, \mathcal{W}_k) = \sum_{j=1}^{r} \|\mathcal{V}_{(:,:,j)} - \mathcal{W}(:,:,j)\mathcal{Y}_{(:,:,j)}\|_F^2 \tag{21}$$

Lines 5–8 of Algorithm 1 imply:

$$\mathcal{L}(\mathcal{Y}, \mathcal{V}, \mathcal{W}_k) - \mathcal{L}(\mathcal{Y}, \mathcal{V}, \mathcal{W}_{k+1}) = \mathcal{L}(\mathcal{Y}, \mathcal{V}, \mathcal{W}_k) - \min_{m} \inf_{\operatorname{rank}(B)=1} \mathcal{L}(\mathcal{Y}, \mathcal{V}, \mathcal{W}_{(m),k} + B)$$

$$\geq \mathcal{L}(\mathcal{Y}, \mathcal{V}, \mathcal{W}_k) - \inf_{\operatorname{rank}(B)=1} \mathcal{L}(\mathcal{Y}, \mathcal{V}, \mathcal{W}_{(1),k} + B) \tag{22}$$

Let's define $B = \alpha C$ where $\alpha \in \mathbb{R}, \operatorname{rank}(C) = 1$, and $\|C\|_2 = 1$. We expand the right hand side of Eq. (22) and write:

$$\mathcal{L}(\mathcal{Y}, \mathcal{V}, \mathcal{W}_k) - \mathcal{L}(\mathcal{Y}, \mathcal{V}, \mathcal{W}_{k+1}) \geq \sup_{\alpha, C:\operatorname{rank}(C)=1, \|C\|_2=1} 2\alpha\langle C\boldsymbol{Y}, \boldsymbol{V} - \mathcal{W}_{(1),k}\boldsymbol{Y}\rangle - \alpha^2\|C\boldsymbol{Y}\|_F^2,$$

where $\boldsymbol{Y}$ and $\boldsymbol{V}$ are used for denoting the matrix created by repeating $\mathcal{Y}_{(:,:,j)}$ and $\mathcal{V}_{(:,:,j)}$ on the diagonal blocks of a block diagonal matrix, respectively. Since the algorithm finds the optimal $B$, we can maximize it with respect to $\alpha$ which yields:

$$\mathcal{L}(\mathcal{Y}, \mathcal{V}, \mathcal{W}_k) - \mathcal{L}(\mathcal{Y}, \mathcal{V}, \mathcal{W}_{k+1}) \geq \sup_{C:\operatorname{rank}(C)=1, \|C\|_2=1} \frac{\langle C\boldsymbol{Y}, \boldsymbol{V} - \mathcal{W}_{(1),k}\boldsymbol{Y}\rangle^2}{\|C\boldsymbol{Y}\|_F^2}$$

$$\geq \sup_{C:\operatorname{rank}(C)=1, \|C\|_2=1} \frac{1}{\sigma_{\max}(\boldsymbol{Y})^2}\langle C\boldsymbol{Y}, \boldsymbol{V} - \mathcal{W}_{(1),k}\boldsymbol{Y}\rangle^2$$

$$= \sup_{C:\operatorname{rank}(C)=1, \|C\|_2=1} \frac{1}{\sigma_{\max}(\boldsymbol{Y})^2}\langle C, (\boldsymbol{V} - \mathcal{W}_{(1),k}\boldsymbol{Y})\boldsymbol{Y}^\top\rangle^2$$

$$= \frac{\sigma_{\max}\left((\boldsymbol{V} - \mathcal{W}_{(1),k}\boldsymbol{Y})\boldsymbol{Y}^\top\right)^2}{\sigma_{\max}(\boldsymbol{V})}$$

Define the residual $R_k = \mathcal{L}(\mathcal{Y}, \mathcal{V}, \mathcal{W}_k) - \mathcal{L}(\mathcal{Y}, \mathcal{V}, \mathcal{W}^*)$. Note that $-(\boldsymbol{V} - \mathcal{W}_{(1),k}\boldsymbol{Y})\boldsymbol{Y}^\top$ is the gradient of the residual function with respect to $\mathcal{W}_{(1),k}$. Since the operator norm and the nuclear norms are dual of each other, using the properties of dual norms we can write for any two matrices $A$ and $B$

$$\langle A, B\rangle \leq \|A\|_2\|B\|_* \tag{23}$$

Thus, using the convexity of the residual function, we can show that

$$R_k - R_{k+1} \geq \frac{\left(\left\|\nabla_{\mathcal{W}_{(1),k}} R_k\right\|_2 \|\mathcal{W}_{(1),k} - \mathcal{W}_{(1)}^*\|_*\right)^2}{\sigma_{\max}(\boldsymbol{Y})^2\|\mathcal{W}_{(1),k} - \mathcal{W}_{(1)}^*\|_*^2} \tag{24}$$

$$\geq \frac{R_k^2}{\sigma_{\max}(\boldsymbol{Y})^2\|\mathcal{W}_{(1),k} - \mathcal{W}_{(1)}^*\|_*^2} \tag{25}$$

$$\geq \frac{R_k^2}{\sigma_{\max}(\boldsymbol{Y})^2\|\mathcal{W}_{(1)}^{*2}\|_*^2} \tag{26}$$

The sequence in Eq. (26) converges to zero according to the following rate [22, Lemma B.2]

$$R_k \leq \frac{(\sigma_{\max}(\boldsymbol{Y})\|\mathcal{W}_{(1)}^*\|_*)^2}{(k+1)}$$

The step in Eq. (25) is due to the fact that the parameter estimation error decreases as the algorithm progresses. This can be seen by noting that the minimum eigenvalue assumption ensures strong convexity of the loss function. $\square$

## C Convex relaxation with ADMM

A convex relaxation approach replaces the constraint $\text{rank}(\mathcal{W}_{(n)})$ with its convex hull $\|\mathcal{W}_{(n)}\|_*$. The mixture regularization in [23] assumes that the $N$-mode tensor $\mathcal{W}$ is a mixture of $N$ auxiliary tensors $\{\mathcal{Z}^n\}$, i.e., $\mathcal{W} = \sum_{n=1}^{N} \mathcal{Z}^n$. It regularizes the nuclear norm of only the mode-$n$ unfolding for the $n$ th tensor $\mathcal{Z}^n$, i.e, $\sum_{n=1}^{N} \|\mathcal{Z}_{(n)}^n\|_*$. The resulting convex relaxed optimization problem is as follows:

$$\widehat{\mathcal{W}} = \underset{\mathcal{W}}{\text{argmin}} \left\{ \mathcal{L}(\mathcal{W}; \mathcal{Y}, \mathcal{V}) + \lambda \sum_{n}^{N} \|\mathcal{Z}_{(n)}^n\|_* \quad \text{s.t.} \quad \sum_{n}^{N} \mathcal{Z}^n = \mathcal{W} \right\} \tag{27}$$

We adapt Alternating Direction Methods of Multiplier (ADMM) [10] for solving the above problem. Due to the coupling of $\{\mathcal{Z}^n\}$ in the summation, each $\mathcal{Z}^n$ is not directly separable from other $\mathcal{Z}^{n'}$. Thus, we employ the coordinate descent algorithm to sequentially solve $\{\mathcal{Z}^n\}$. Given the augmented Lagrangian of problem as follows, the ADMM-based algorithm is elaborated in Algo. 2.

$$F(\mathcal{W}, \{\mathcal{Z}^n\}, \mathcal{C}) = \mathcal{L}(\mathcal{W}; \mathcal{Y}, \mathcal{V}) + \lambda \sum_{n=1}^{N} \|\mathcal{Z}_{(n)}^n\|_* + \frac{\beta}{2} \sum_{n} \|\mathcal{W} - \sum_{n} \mathcal{Z}^n\|_F^2 - \langle \mathcal{C}, \mathcal{W} - \sum_{n=1}^{N} \mathcal{Z}^n \rangle \tag{28}$$

---

**Algorithm 2** ADMM for solving Eq. (6)

1: **Input:** transformed data $\mathcal{Y}, \mathcal{V}$ of $M$ variables, hyper-parameters $\lambda, \beta$.
2: **Output:** $N$ mode tensor $\mathcal{W}$
3: Initialize $\mathcal{W}, \{\mathcal{Z}^n\}, \mathcal{C}$ to zero.
4: **repeat**
5:     $\mathcal{W} \leftarrow \text{argmin}_{\mathcal{W}} \left\{ \mathcal{L}(\mathcal{W}; \mathcal{Y}, \mathcal{V}) + \frac{\beta}{2} \|\mathcal{W} - \sum_{n=1}^{N} \mathcal{Z}^n - \mathcal{C}\|_F^2 \right\}$.
6:     **repeat**
7:        **for** variable $n = 1$ **to** $N$ **do**
8:           $\mathcal{Z}_{(n)}^n = \text{shrink}_{\frac{\lambda}{\beta}} \left( \mathcal{W}_{(n)} - \frac{1}{\beta}\mathcal{C} - \sum_{n' \neq n} \mathcal{Z}_{(n')}^{n'} \right)$.
9:        **end for**
10:    **until** solution $\{\mathcal{Z}^n\}$ converge
11:    $\mathcal{C} \leftarrow \mathcal{C} - \beta(\mathcal{W} - \sum_{n=1}^{N} \mathcal{Z}^n)$.
12: **until** objective function converges

---

The sub-routine $\text{shrink}_\alpha(A)$ applies a soft-thresholding rule at level $\alpha$ to the singular values of the input matrix $A$. The following lemma shows the convergence of ADMM-based solver for our problem.

**Lemma 3.** *[3] For the constrained problem* $\min_{x,y} f(x) + g(y), \text{s.t} \quad x \in C_x, y \in C_y, Gx = y$, *If either $\{C_x, C_y\}$ are bounded or $G'G$ is invertible, and the optimal solution set is nonempty. A sequence of solutions $\{x, y\}$ generated by ADMM is bounded and every limit point is an optimal solution of the original problem.*

## D Derivation of the unified formulation

In this section, we demonstrate how we can use Eq. (6) to solve Eqs. (4) and (5). In the cokriging problem, it is easy to see that with $\mathcal{Y}_{:,:,m} = H$ and $\mathcal{V}_{:,:,m} = \mathcal{X}_{\Omega,m}$ for $m = 1, \ldots, M$ the problems are equivalent. In the forecasting problem, $H$ is full rank and the mapping defined by $\mathcal{W} \mapsto \tilde{\mathcal{W}}$ : $\tilde{\mathcal{W}}_{:,:,m} = H\mathcal{W}_{:,:,m}$ for $m = 1, \ldots, M$ preserves the tensor rank, i.e., $\text{rank}(\mathcal{W}) = \text{rank}(\tilde{\mathcal{W}})$. This suggests that we can solve Eq. (4) as follows: first solve Eq. (6) with $\mathcal{Y}_{:,:,m} = \mathbf{X}_{K+1:T,m}$ and $\mathcal{V}_{:,:,m} = \mathcal{X}_{:,:,m}$ and obtain its solution as $\tilde{\mathcal{W}}$; then compute $\mathcal{W}_{:,:,m} = H^{-1}\tilde{\mathcal{W}}_{:,:,m}$.