[Reviews · NeurIPS 2014]

Submitted by Assigned_Reviewer_12

The authors consider the task of interpolation (cokriging) and forecasting on spatio-temporal tensor data. They show that both tasks can be posed as a (non-convex) low-rank tensor optimization problem, and proceed to present a learning method that combines a forward greedy selection with an optional orthogonal projection step. Experiments consider simulated data, two climatological datasets, and another containing FourSquare checkins.

I would not consider myself an expert in this particular domain, and am not in a position to comment on novelty or the presented proofs. That said, I find the paper clearly written. The presented greedy approach to low-rank learning seems reasonable, and the experiments show clear gains relative to other methods. In all this seems a solid paper with no obvious faults.

Minor Points

- While the discussion in 2.1 and 2.2 is easy to follow, it is not obvious to me how equation (4) in section 2.3 is equivalent to equation (1), how equation (5) is equivalent to (3), and how both (4) and (5) are equivalent to (6). A reference or more verbose explanation in the supplementary material would be helpful.

- When discussing Fig 1c the authors write that the "run time of ADMM increase[s] rapidly with data size while the greedy algorithm stays steady". This does not seem an accurate representation, as the run time clearly increases for all algorithms. If the longest performed run is indeed only 1000 seconds, then I would encourage the authors to further test the scalability of their algorithm with additional longer runs.

- Is the RMSE used to characterize cokriging and forecasting performance normalized in some way? If so, it would be interesting if the authors could provide some insight into why there are comparatively large gains in the CCDS dataset, whereas all algorithms appear to perform well on the Foursquare dataset

- Line 34: From machine learning perspective -> From [a] machine learning perspective

- Line 305: five folds cross-validation -> five-fold cross-validation
Summary: The greedy approach to low-rank learning presented in this paper looks reasonable, and the experiments show clear gains relative to other methods. In all this seems a solid, well-written, paper with no obvious faults.

Submitted by Assigned_Reviewer_21

A multivariate spatio-temporal analysis based on cokriging and forecasting is presented. The analysis is based on the low rank tensor learning framework. An efficient greedy learning algorithm is implemented. Results obtained both from synthetic and real data problems are discussed.
Summary: I have just a couple of minor comments.
1) there is a typo at page 2, between section 2 and subsection 2.1: of patio-temporal
2) I would prefer "subject to" in place of "s.t." (formulae 5 and 6)

Submitted by Assigned_Reviewer_29

Authors present a unified low rank tensor learning framework for two main tasks in multivariate spatio-temporal analysis: cokriging and forecasting. Authors present an efficient greedy algorithm that can be used for both tasks.

Positives:
* Paper is well written and has good mathematical foundation of the approach taken to solve the tasks.
* Positive results are shown on both synthetic and real-world data when compared to other algorithms in the literature.

Areas of improvement:
* It is unclear how different tuning parameters are chosen for different algorithms when applied to different data.
* It is unclear if this approach will work if the data had more than 3 dimensions.
* Multiple fold cross validation results are shown in Table 1, 2 without any confidence intervals.
* Can the forecasting approach be taken one step further to classify anomalies in climate data such as hurricance, drought, etc?
* Table 2 shows Tucker beats Ortho alg on USHCN data. Why? Are the results significant? Add discussion on what this means.

Few typos:
- Pg. 1 Oftentimes --> Often times
- Pg. 1, 7 five folds cross validation --> five-fold cross validation
Summary: Paper has good mathematical foundation for learning low rank tensors from spatio-temporal data. I would have loved to see discussion on why cokriging and forecasting are important especially for the datasets tested in the paper.
Author Feedback
Author rebuttal: Thank you for the insightful comments and positive feedback.

***Response to Reviewer #12***

- Normalized RMSE: The time series are normalized to mean zero and unit variance before experiments, thus RMSE values are normalized.

We will perform more experiments to extend the X-axis for the scalability experiment.

We will provide the details of derivation of Eq. (4-6) in the supplementary materials.

***Response to Reviewer #29***

- Generalization to higher dimensional tensors: This is an interesting point! We believe this method should work for tensors with more than 3 modes. While in Algorithm 1, there is no constraint on the number of the modes, the proof of Theorem 1 (specifically, lines 550-555) suggests that the bound also should hold for tensors with N > 3 modes if an appropriately defined loss function is given. Detailed discussion will be provided on the generalization.

- Tuning parameters: the Laplacian penalty parameter \mu controls the local consistency strength; the termination criteria \eta makes trade-off between the approximation quality by low-rank and run time. These parameters are tuned via cross-validation. The kernel bandwidth is tuned on a subset of the data using cross-validation. In practice, we rescale the \Delta variables by the objective function in the beginning to make it unitless and thus making the tuning process across different datasets easier.

- Confidence intervals: The confidence intervals will be provided.

- Anomaly detection: Thank you for pointing this out. This could be an exciting future work.

- USHCN data: While Tucker decomposition slightly outperforms orthogonal greedy with respect to forecasting accuracy, it is much slower than our algorithm. We will investigate and provide a discussion for the underlying reason.